# Right Ventricular Morphology and Function after Exercise Training in People with Systemic Sclerosis: A Randomized Controlled Pilot Study

**DOI:** 10.3390/life13020545

**Published:** 2023-02-15

**Authors:** Maria Anifanti, Andriana Teloudi, Alexandros Mitropoulos, Niki Syrakou, Eleni Pagkopoulou, Eva Triantafyllidou, Carina Boström, Louise Pyndt Diederichsen, Giovanna Cuomo, Theodoros Dimitroulas, Markos Klonizakis, Evangelia Kouidi

**Affiliations:** 1Laboratory of Sports Medicine, Department of Physical Education and Sports Science, Aristotle University of Thessaloniki, 57001 Thessaloniki, Greece; 2Lifestyle, Exercise and Nutrition Improvement (LENI) Research Group, Department of Nursing and Midwifery, Sheffield Hallam University, Sheffield S10 2BP, UK; 34th Department of Internal Medicine, School of Medicine, Hipokration Hospital, Aristotle University of Thessaloniki, 54642 Thessaloniki, Greece; 4Department of Neurobiology, Care Sciences and Society, Karolinska Institutet, SE-104 35 Stockholm, Sweden; 5Copenhagen Research Centre for Autoimmune Connective Tissue Diseases, Copenhagen University, Rigshospitalet, DK-1165 Copenhagen, Denmark; 6Department of Precision Medicine, University of Campania L. Vanvitelli, 80138 Naples, Italy

**Keywords:** systemic sclerosis, exercise training, right ventricular function, cardiopulmonary exercise testing, echocardiography

## Abstract

Background: Vascular dysfunction and its concomitant multi-organ involvement, including cardiac involvement, affects prognosis in systemic sclerosis (SSc) patients. Regular exercise has demonstrated to be able to improve vascular function in SSc. However, the effects of an exercise program on the heart and specifically in right ventricular (RV) morphology and function in SSc have yet to be explored. The study aimed to examine whether a 3-month combined exercise program can affect RV morphology and function in SSc patients. Methods: Twenty-eight SSc patients were randomly allocated to either the exercise training (ET) or the control (CON) group. Baseline and follow-up assessments consisted of a cardiopulmonary exercise test along with both a conventional and a two-dimensional speckle tracking echocardiography (2DSTE) focused on RV morphology and function. Following the baseline assessments, Group ET participated in a supervised combined exercise program for 12 weeks, while group CON received their usual care. Results: The ET group demonstrated increases in peak oxygen consumption by 25.1% (*p* < 0.001), global RV free wall longitudinal systolic strain by 6.69% (*p* < 0.03), RV free wall longitudinal systolic strain of the basal segment by 13.5% (*p* < 0.001), and global RV four-chamber longitudinal systolic strain by 6.76% (*p* < 0.03) following the exercise program. No differences were observed in group CON. Conclusions: Combined exercise improved cardiorespiratory efficiency and indices of RV systolic function, as assessed by the 2DSTE, in SSc patients.

## 1. Introduction

Systemic sclerosis (SSc) involves significant vascular and cardiac complications which severely affect SSc patients’ quality of life [1]. The cardiac complications include both functional and structural heart abnormalities [2]. SSc-related cardiac disorders include the development of coronary artery disease, pulmonary arterial hypertension (PAH), cardiac autonomic dysfunction with rhythm and conduction disturbances, myocarditis, pericarditis, valvular disease, and heart failure [3]. The development of heart failure and arrhythmias was found to significantly affect survival [4].

In SSc patients, the presence of left ventricular (LV) remodeling and dysfunction are well-studied to date [5]. In particular, studies which used a two-dimensional speckle tracking echocardiography (2DSTE) were able to describe cardiac deformation and detect early contraction abnormalities [3,6]. 2DSTE was able to reveal impaired RV free wall contractility patterns in SSc which were not detected by standard 2D echocardiographic measures [7].

Exercise training in SSc can induce favorable effects on microvascular function and prevent debilitating clinical manifestations, such as digital ulcers [8,9]. However, the cardiac morphology and function after exercise training in SSc patients have not been studied yet. Echocardiographic indices of both LV [10] and RV [11] function were found to be correlated with SSc patients’ cardiorespiratory fitness, as assessed by peak oxygen consumption (VO_2_peak). Although this relationship is of high importance, no study has examined whether an improvement of VO_2_peak caused by exercise training is associated with improved cardiac function. The primary objective of the present study was to assess possible exercise-induced changes in RV morphology and function with the use of conventional 2D echocardiograms and 2DSTE. A secondary objective was to explore the relationship between VO_2_peak and RV echocardiographic indices after exercise training in SSc patients.

## 2. Methods

### 2.1. Study Participants

Patients with SSc from the Fourth Department of Internal Medicine, Hippokration University Hospital, Thessaloniki, Greece were invited to participate in the study. Patients with SSc in a stable clinical status, aged over 18 years, without injuries, active digital ulcers, or other contraindications to exercise training were eligible. Patients with unstable angina, uncontrolled arterial hypertension, advanced heart failure, mental or cognitive impairment, or who were already participating in a structured exercise training program were excluded from the study. After reviewing their medical records, detailed information about the study objectives and protocol was given to all eligible volunteers. Thereafter, the SSc patients provided informed consent. The clinical study was reviewed and approved by the Research Ethics and Deontology Committee of the Aristotle University of Thessaloniki, Greece (Prot. No.: 176497/2021). This is part of a large European multi-center randomized controlled trial entitled “Exploring the effects of a combined exercise program on pain and fatigue outcomes in people with systemic sclerosis” [12], which is registered on ClinicalTrials.gov (NCT05234671), accessed on 9 January 2023.

### 2.2. Study Protocol

Baseline assessment included the patient’s medical history (i.e., history of digital ulcers, comorbidities, and medical treatment), anthropometric characteristics, a cardiac screening including an electrocardiogram and an echocardiographic assessment, as well as an arm crank cardiopulmonary exercise testing (CPET) evaluation. After initial assessment, all volunteers were allocated to either the exercise (ET) or the control (CON) group at random, based on three factors: (i) SSc type, (ii) SSc duration in yrs, and (iii) SSc severity based on the modified Rodnan skin score and the Medsger severity score. The group allocation was conducted by an independent statistician blinded to the study’s procedures (e.g., baseline assessments) using a computer program. The ET group (n = 14; 71.4% female) followed a supervised combined exercise training program for 12 weeks, while the CON group (n = 14; 100% female) received only standard treatment and abstained from any structured exercise program for the same period. All participants were reassessed at the end of the three-month study using the same tests. Rheumatologists, outcome assessors, and researchers collecting and analyzing the patients’ data were blinded to group allocation.

### 2.3. CPET via Arm Ergometry

At the beginning of the study and three months later, all of the SSc patients underwent maximal cardiopulmonary exercise testing using an arm ergometer (Monark 881E Rehab Trainer, MONARK EXERCISE AB, Vansbro, Sweden). The Med Graphics Breeze Suite CPX Ultima ergospirometer (Medical Graphics Corp, MN) was used to measure and analyze the expiratory gases. An incremental graded protocol was used until volitional exhaustion. After a three-minute warm up, males commenced the test at 30 W and females commenced the test at 20 W with a pedal speed of 70 revolutions per minute. The workload increased by 10 W/min for males and 6 W/min for females. [8,13]. At the end of the test, patients had an active recovery period (for 3 min at a low rate without any load) and a three-minute passive recovery period. There was continuous electrocardiographic monitoring during the test, while the blood pressure measurement was obtained at the beginning and the end of the test and recovery (i.e., 10 min post exercise termination). VO_2_peak was recorded as the average VO_2_ during the last 30 sec of the exercise test.

### 2.4. Echocardiographic Measurements

A Vivid S70 (GE Medical; Horten, Norway) equipped with an M5S phased-array transducer was used. All images were stored in EchoPAC, version 204, and later analyzed by two cardiologists, who were blinded to group assignment and time period (either baseline or end of study). All analyses were performed according to the European Association of Cardiovascular Imaging and American Society of Echocardiography guidelines [14,15]. The LV measurements were obtained from the parasternal long axis view and from four- and two-chamber views. The LV end-diastolic diameter (LVEDD), the LV interventricular septum thickness (LVIVSd), and the LV posterior wall thickness (LVPWd) were obtained from the parasternal long-axis view at the end-diastolic phase at the level of the mitral valve leaflet tips. The maximum volume of the left atrium (LAV) was measured using the biplane disk summation technique at the end-systole and indexed using BSA (LAVi). The modified biplane Simpson’s method was used to calculate the LV end diastolic volume (LVEDV), stroke volume (SV), cardiac output (CO), and LV ejection fraction (LVEF). The diastolic function and LV filling pressure were assessed using early diastolic transmitral flow velocity (MVE), late diastolic transmitral flow velocity (MVA), the ratio (MVE/A) of the two, and the average E/E’ ratio. Pulsed-wave TDI E’ velocity was obtained from the apical four-chamber view, at the lateral and septal basal regions, and based on this the average E’ and average E/E’ ratio were computed.

RV linear dimensions were assessed using two acoustic windows. Proximal RV outflow diameter (RVOT prox) was measured from the parasternal long axis and the basal RV dimension (RV bas) was measured from the RV-focused view at the end-diastolic phase. Simpson’s method was used to calculate the right atrial volume (RAVol). The tricuspid annular plane systolic excursion (TAPSE) was measured using M-mode echocardiography between the end-diastole and peak systole. The RV diastolic function was assessed using early diastolic transtricuspid flow velocity (TVE), late diastolic transtricuspid flow velocity (TVA), and the ratio of the two (TVE/A). Peak systolic velocity (TVS’), peak early diastolic velocity (TVE’), and peak late diastolic velocity (TVA’) of the tricuspid annulus were obtained using the apical approach with pulsed-wave TDI. The simplified Bernoulli equation was used to calculate the pulmonary artery systolic pressure (PASP) from the peak TR velocity, according to the guidelines. RV acceleration time (RVAT) was obtained from the pulmonary flow using a PW Doppler.

RV longitudinal systolic strain (RVLS) obtained via 2DSTE derived from an RV-focused apical view [16,17]. The record of images obtained had a temporal resolution of 50–80 fps at end-expiration. The endocardial contour was manually traced. The peak longitudinal systolic strain from the basal (RVFWLSbas), mid (RVFWLSmid), and apical (RVFWLSap) segments of the RV free wall were obtained. Global RV free wall longitudinal systolic strain (RVFWLS) was calculated as the average value of the longitudinal strains of the three free wall RV segments. Moreover, global RV four-chamber longitudinal systolic strain (RV4CLS) was calculated as the average value of the longitudinal strains of the six segments of the RV free wall and interventricular septum view [17].

### 2.5. Exercise Training Program

After the initial assessments, the patients of Group ΕΤ attended two combined (aerobic and resistance) training sessions per week for three months [9]. Each session was executed as a one-to-one supervised session by a qualified exercise trainer with experience with SSc patients. During the workouts, both heart rate and perceived exertion (Borg’s scale) were measured at the end of each part of the session. The aerobic exercise training part was performed using a Monark 881E Rehab Trainer. Each session started with a warm up, then the patients performed light arm cranking, and the session ended with a cool-down, with each part lasting 5 min. The main exercise training program lasted 30 min and consisted of high-intensity interval training of 30:30 (cycling for 30 s at the Wmax achieved during CPET following by 30 s passive recovery) on the arm ergometer. Thereafter, patients performed 15 to 30 min of resistance exercise training, which included five exercises with 10 repetitions each for three circuits at 75–80% of the one repetition maximum. Specifically, the exercises were: a. shoulder lateral raise, b. bicep curl, c. triceps extension, and d. hand squeeze of a handgrip dynamometer, all performed in a sitting position, as well as e. chest press on a bench in a 30° supine position. There was a 2–3 min rest interval between circuits.

### 2.6. Statistics

Statistical analyses were performed using the SPSS 25.0 software for Windows (Chicago, IL, USA). The Shapiro–Wilk test was used to examine whether the variables were normally distributed. The T-test was used to evaluate the initial differences between the two groups and a chi-square (Χ²) test was used for the categorical data. Two-way repeated measures ANOVA with the Bonferroni post-hoc test was used to compare the mean differences within groups and between the two groups. Moreover, the Pearson correlation coefficient was used to assess the correlation between variables at baseline and the end of the study. *p*-values ≤ 0.05 were considered statistically significant.

## 3. Results

Twenty-eight patients with SSc participated in the study (24 females, mean age 57.21 ± 10.76 years). All patients in group EΤ participated in at least 90% of the provided training sessions, and thus were analyzed. There was no adverse effect from the exercise training program and no dropouts during the three-month study (Figure 1). The clinical characteristics of the participants are presented in Table 1. No patient had diabetes mellitus and kidney or liver disease. At the beginning of the study, VO_2_peak values did not differ significantly between the two groups (*p* = 0.79). After three months, VO_2_peak increased by 25.1% (from 13.9 ± 4.5 to 17.4 ± 3.9 mL/kg/min, *p* < 0.001) in the ΕΤ group and decreased by 12.8% (from 14.0 ± 2.6 to 12.2 ± 2.6 mL/kg/min *p* < 0.03) in the CON group, reaching a 29.8% difference between the two groups (*p* = 0.001).

There was no baseline difference observed in the echocardiographic indices studied between the two groups. From the baseline echocardiographic study, all patients had normal LV and LA sizes, SV and CO, and systolic and diastolic function of the LV before and after the three-month intervention (Table 2). Similarly, measurements of RV dimensions, RVOT prox, and RVbas were normal in size (Table 3). RAVol was within the normal range. Moreover, conventional parameters of RV function (TAPSE, TVS’) and diastolic function of the RV (TVE/A, TVE’) were normal in both groups. The patients did not have PAH according to the assessment of PASP and RVAT, since both parameters were within normal ranges. All LV parameters remained unchanged at the end of the three-month study.

At baseline, all measurements of RV longitudinal strain were lower in both groups in comparison to established reference values and there was no between-group difference [17,18]. After the three-month exercise intervention in group ET, RVFWLS increased by 6.69% (*p* < 0.03), RVFWLSbas increased by 13.5% (*p* < 0.001) and RV4CLS increased by 6.76% (*p* < 0.03) (Figure 2). Patients in the CON group did not show any statistically significant difference in the parameters studied over time.

After three months, there were statistically significant differences in RVFWLSbas of 14.2% (*p* = 0.001), RVFWLS of 7.17% (*p* = 0.04), and RV4CLS of 7.2% (*p* = 0.05) between the two groups.

At the end of the study, a significant negative linear correlation was found between VO_2_peak and RVFWLS (r = −0.882, *p* < 0.001), RV4CLS (r = − 0.766, *p* = 0.001), and RVFWBAS (r = − 0.822, *p* < 0.001) in group ET (Figure 3).

## 4. Discussion

The results of the 2DSTE in the current study indicated that both the global and regional RVLS were diminished in our study’s participants, suggesting RV systolic dysfunction which was not detected by the conventional echocardiographic study. In addition, following the exercise program, the RVFWLS bas segment, global RVFWLS, and RV4CLS were significantly improved in the exercise group.

Our baseline echocardiographic study supports the need to evaluate RV function using novel echocardiographic techniques in SSc patients. In our asymptomatic patients with normal ranges of RV function and without PAH according to standard echocardiographic imaging, 2DSTE revealed lower absolute values in all RV free wall segments compared to the proposed reference values [18], indicating diminished deformation along the longitudinal axis of the RV. Similarly, several previous studies have pointed out the need to regularly monitor cardiac function using 2DSTE in clinical practice even in SSc patients without known cardiac disease and PAH [7,19,20].

However, comparing our results with those derived from previous studies that have analyzed the RV wall deformation in SSc patients with normal PASP levels, there are several discrepancies. Some studies that have examined the RV function in SSc patients using 2DSTE have reported that the strain values measured at the level of either the basal RVFW [19] or the global RVFWLS [20,21] in the SSc patients did not differ from the healthy controls. Moreover, the basal RVFWLS was found to be increased in the SSc patients compared to the healthy controls, unlike our findings [7]. Consistent with our results, Mukherjee et al. [7] reported impaired RVFWLS in the apex and mid segments and global RVLS in SSc patients compared to the reference values of RV strain. These findings are suggestive of subclinical RV systolic dysfunction detected early. It is well-known that there is an association of longitudinal strain and afterload. Our results can be attributed to an increased afterload derived by increased PASP, which can cause a reduction in global and regional RVLS suggestive of RV dysfunction. Since our patients did not have PAH, it seems that small or moderate increases in PASP can affect RV systolic function over time. Considering that the pathophysiology of SSc is expressed with microvascular damage, inflammation and pulmonary fibrosis, pulmonary involvement, and increases of RV afterload, early detection is important for prognostication [22].

Apart from cardiac and pulmonary involvement, vascular injury and consequent tissue fibrosis also affects skeletal muscle morphology and function [23,24]. Consequently, patients present pain, muscle weakness, and fatigue, which reduce their daily physical activity and lead to functional disability [25]. Ross et al. [26] have shown that there is an association between skeletal muscle oedema and low exercise capacity in SSc patients. The baseline CPET assessment using an arm ergometer showed that our patients had reduced aerobic capacity. Similar to our results, several studies have found that even SSc patients without severe cardiopulmonary disease have reduced exercise capacity [26,27]. Consequently, they choose a sedentary lifestyle [28]. The vicious cycle of inactivity worsens their physical function and affects their quality of life (HRQoL) [29]. Exercise capacity was found to have a prognostic value in SSc patients [30].

To date, several studies have demonstrated that regular exercise in SSc patients can improve their disease symptoms, exercise capacity, physical function in daily life, and psychosocial status [31,32]. Thus, exercise is highly recommended [33,34]. In particular, aerobic and resistance exercise training programs were found to reduce cardiac risk factors and improve patients’ HRQL [25,35]. Moreover, we have shown that a 12-week high intensity interval combined exercise training program led to significant improvements in vascular function in SSc patients [35]. We have used the same combined exercise protocol to examine the exercise-induced changes in RV function in SSc patients. Our results show that a 12-week combined exercise program can improve aerobic capacity and indices of RV systolic function in patients with subclinical RV systolic dysfunction. Specifically, there were increases in RVFWLSbas, RVFWLS, and RV4CLS, which were related to VO_2_peak after training in our exercised SSc patients. Similarly, a recent study has shown that the TAPSE/sPAP ratio, which is an echocardiographic parameter of RV function, was correlated with VO_2_max [11]. LV global longitudinal and circumferential strains were also found to be independently associated with VO_2_peak [10]. As indicated from our findings (Table 2), the VO_2_peak improvement is accounted for by peripheral muscle adaptations rather an improved stroke volume and cardiac output following high-intensity interval training [36]. In addition to aerobic exercise, resistance training has been demonstrated to improve VO_2_peak, explained primarily by a change in the arteriovenous oxygen difference (a-VO_2_diff), rather than in SV, in older adults [37]. An enhanced capillary density and myoglobin concentration of muscle account for the increase in a-VO_2_diff [38], as well as an increase in muscle mitochondria content and enzyme activity [39]. Moreover, adaptations in peripheral vascular resistance, leading to an increased muscular blood flow in working muscle, may contribute to the increase in VO_2_max following RT.

Microcirculatory adaptations have also contributed to the RV function improvement following exercise in our study. Namely, a previous study by our research group has demonstrated that this exercise training program was able to improve both the endothelial-dependent microvascular function and transcutaneous oxygen pressure at the thoracic level in SSc patients [9], demonstrating the systemic effects of exercise on microvascular function. This systemic exercise-induced microvascular improvement may also contribute to the improved RV function in our exercised patients. Moreover, a recent study using supine bicycle echocardiography in SSc patients has shown that acute exercise load can increase the reduced mid and apical RVLS at rest, suggestive of RV contractile reserve [40].

The control group presented with a decline in VO_2_peak at the follow-up assessments. As described above, the short-term increase or decrease in VO_2_peak is not related to the cardiac output (function and structure), but rather to peripheral muscular and vascular changes. We know that muscular deconditioning can be accelerated with increased physical inactivity (which is dependent on various factors such as socioeconomic, environmental, psychological, and health-related changes such as disease exacerbation). At the same time, a microvascular disease progression is observed over time in SSc patients [8]. More specifically, those who took part in the control group (i.e., no exercise intervention) presented an incidence of digital ulcers of up to 36% compared to 0% in the exercise group. In the same study [8], vascular function declined in the control group as opposed to in the exercise group, in which it was improved. Therefore, muscular deconditioning, as well as disease-induced vascular progression, could explain the decline in VO_2_peak in the control group in the present study.

A limitation of the study is the small sample size and the absence of known cardiac disease and PAH, and thus we cannot generalize our results. Second, there was no healthy control group and the proposed reference values were used for comparison. Third, there was no long-term follow-up to examine the potential prognostic impact of exercise training; nevertheless, from our previous studies [9], we have demonstrated that both VO_2_peak and microvascular adaptations return to baseline values at six-month follow up. Finally, our study presented a group imbalance concerning the type of SSc; namely, the exercise group included more participants with diffuse SSc, and this could have affected our results, but there is no specific evidence to support this hypothesis.

## 5. Conclusions

Our results support the significance of 2DSTE in clinical practice for the early detection of cardiac deformation in SSc patients. In conclusion, a combined exercise training program was effective in improving cardiorespiratory efficiency and RV systolic function in SSc patients with subclinical RV dysfunction.

## Figures and Tables

**Figure 1 life-13-00545-f001:**
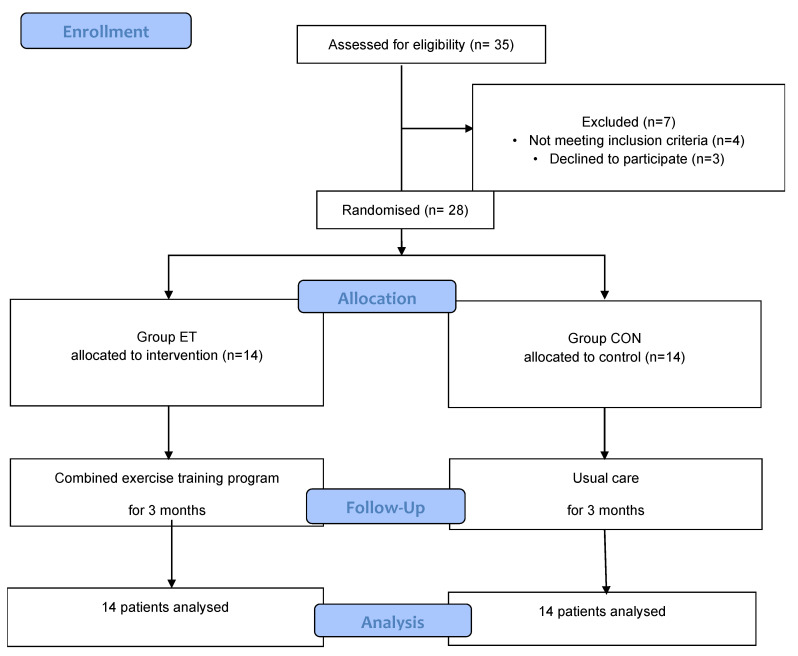
Flowchart of the participants.

**Figure 2 life-13-00545-f002:**
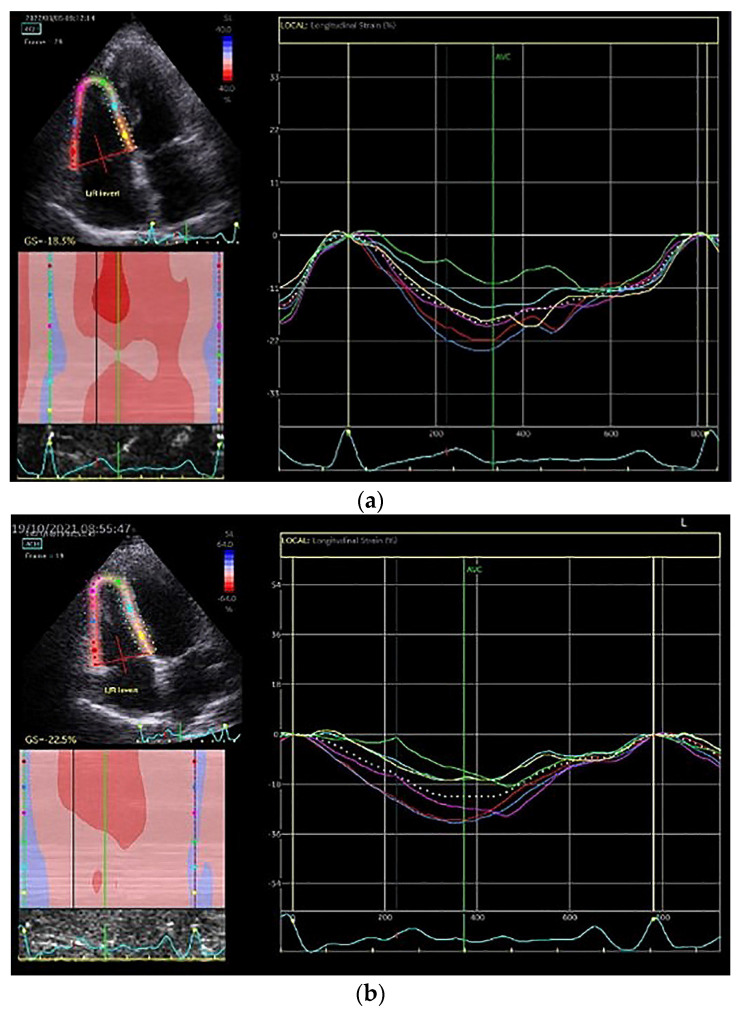
RV strain measurements in an SSc patient (**a**) at baseline (RV4CLS = −18.5%) and (**b**) after exercise training (RV4CLS = −22.5%).

**Figure 3 life-13-00545-f003:**
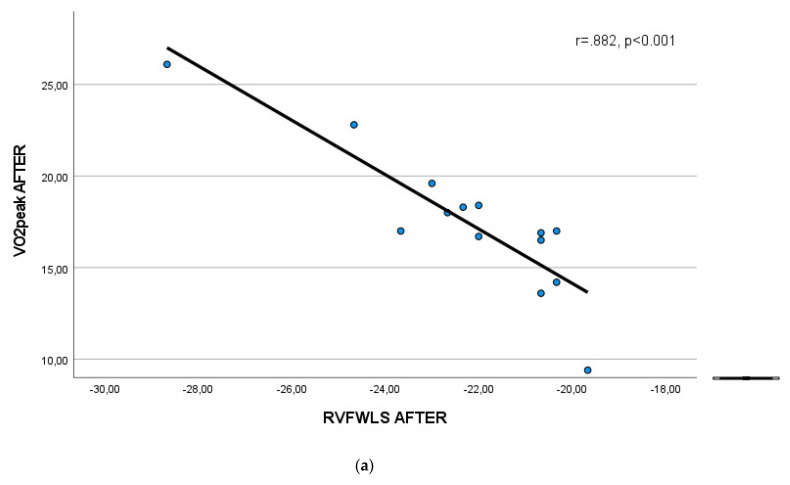
Scatterplots presenting the linear correlation between VO_2_peak and (**a**) RVFWLS, (**b**) RV4CLS, and (**c**) RVFWBAS in group ET at the end of the study.

**Table 1 life-13-00545-t001:** Clinical parameters of the SSc patients at baseline.

Parameters	Total	Group ET	Group CON	*p*
N	28	14	14	
Gender				
Male (n,%)	4 (14.29%)	4 (28.6%)	0 (0%)	0.098
Female (n,%)	24 (85.71%)	10 (71.4%)	14 (100%)	
Age (years)	57.21 ± 10.76	56.14 ± 10.212	58.29 ± 11.56	0.61
BSA	1.71 ± 0.14	1.73 ± 0.16	1.70 ± 0.13	0.54
Disease duration [IQR] (years)	6.50 [10.5]	6.50 [4.8]	8.00 [13.5]	0.80
**Type of disease**				
Limited (n,%)	15 (53.57%)	5 (35.7%)	10 (71.4%)	0.06
Diffuse (n,%)	13 (46.43%)	9 (64.3%)	4 (28.6%)	
**ANA**				
Positive (n,%)	28 (100%)	14	14	1.00
Negative (n,%)	0	0	0	
**ACA**				
Positive (n,%)	8 (28.57%)	2 (14.3%)	6 (42.9%)	0.209
Negative (n,%)	20 (71.43)	12 (85.7%)	8 (57.1%)	
**SCL70**				
Positive (n,%)	13 (46.43%)	9 (64.3%)	4 (28.6%)	0.058
Negative (n,%)	15 (53.57%)	5 (35.7%)	10 (71.4%)	

BSA: body surface area, IQR: interquartile range; ANA: antinuclear antibodies; ACA: Anti-centromere antibodies; SCL70: serum anti-topoisomerase I antibody.

**Table 2 life-13-00545-t002:** Conventional 2D echocardiographic parameters of the LV at the beginning and the end of the study (mean ± SD).

	ET			CON			ET vs. CON *p*	
LV Indices	Baseline	After	*p*	Baseline	After	*p*	Baseline	After
LVIVSd (mm)	7.7 ± 0.9	7.8 ± 0.7	0.33	7.8 ± 0.6	7.9 ± 0.8	0.69	0.65	0.82
LVEDD (mm)	46.1 ± 3.2	46.7 ± 2.4	0.44	46.3 ± 3.4	47.7 ± 2.7	0.10	0.88	0.30
LVPWd (mm)	7.7 ± 0.9	7.6 ± 0.7	0.79	7.7 ± 0.7	7.7 ± 0.8	0.99	0.86	0.78
LVEDV (mL)	84.7 ± 17	86.5 ± 5.9	0.67	82.1 ± 11	81.7 ± 8.1	0.90	0.66	0.08
SV (ml)	52.0 ± 7.6	52.3 ± 7.7	0.68	49.7 ± 7.3	49.5 ± 7.4	0.82	0.21	0.12
CO (l/min)	3.72 ± 0.6	3.77 ± 0.5	0.41	3.80 ± 0.5	3.87 ± 0.5	0.25	0.72	0.67
LVEF (%)	62.7 ± 5.3	63.7 ± 4.6	0.50	62.2 ± 4.3	62.0 ± 3.0	0.86	0.81	0.36
MVE (m/s)	0.75 ± 0.1	0.69 ± 0.1	0.26	0.74 ± 0.08	0.73 ± 0.09	0.73	0.84	0.09
MVA (m/s)	0.69 ± 0.1	0.68 ± 0.1	0.59	0.71 ± 0.1	0.72 ± 0.1	0.94	0.66	0.34
MVE/A	1.11 ± 0.3	1.03 ± 0.2	0.46	1.04 ± 0.1	1.04 ± 0.1	0.99	0.44	0.91
E/E’	6.6 ± 1.7	7.2 ± 1.6	0.44	6.9 ± 1.1	7.3 ± 1.3	0.43	0.66	0.90
LAVi (ml/m^2^)	25.6 ± 7.6	26.6 ± 4.3	0.63	24.7 ± 4.3	25.7 ± 4.5	0.45	0.73	0.65

LVIVSd: LV interventricular septum thickness at end diastole, LVEDD: LV end-diastolic dimension, LVPWd: LV posterior wall thickness at end diastole, LVEDV: LV end-diastolic volume, SV: stroke volume, CO: cardiac output, LVEF: LV ejection fraction, MVE: early diastolic transmitral flow velocity, MVA: late diastolic transmitral flow velocity, MVE/A: ratio of early to late diastolic transmitral flow velocity, E/E’: ratio of early diastolic inflow velocity to early diastolic annular velocity, LAV: left atrial volume.

**Table 3 life-13-00545-t003:** Conventional 2D echocardiographic and 2DSTE Parameters of the RV at the beginning and the end of the study (mean ± SD).

	ET			CON			ET vs. CON *p*	
RV Indices	Baseline	After	*p*	Baseline	After	*p*	Baseline	After
RV bas (mm)	38.7 ± 2.6	37.1 ± 2.9	0.08	37.0 ± 2.1	37.1 ± 2.3	0.89	0.11	0.99
RVOT prox (mm)	30.3 ± 2.7	30.1 ± 2.1	0.77	29.6 ± 1.6	28.4 ± 2.3	0.08	0.46	0.06
RAVol/BSA (mL/m^2^)	21.7 ± 4.2	21.2 ± 3.0	0.49	19.8 ± 6.0	19.2 ± 3.9	0.70	0.40	0.09
TVE (m/s)	0.63 ± 0.07	0.68 ± 0.08	0.16	0.67 ± 0.06	0.64 ± 0.16	0.64	0.14	0.54
TVA (m/s)	0.52 ± 0.06	0.59 ± 0.07	0.06	0.57 ± 0.12	0.57 ± 0.09	0.99	0.07	0.49
TVE/A	1.23 ± 0.19	1.16 ± 0.16	0.32	1.20 ± 0.18	1.13 ± 0.19	0.32	0.68	0.70
TVS’ (m/s)	0.12 ± 0.01	0.12 ± 0.01	0.57	0.12 ± 0.01	0.12 ± 0.01	0.45	0.47	0.52
TVE’ (m/s	0.13 ± 0.02	0.12 ± 0.02	0.34	0.13 ± 0.02	0.13 ± 0.01	0.67	0.94	0.27
TVA’ (m/s)	0.15 ± 0.03	0.14 ± 0.02	0.43	0.15 ± 0.02	0.15 ± 0.02	0.69	0.91	0.08
TAPSE (mm)	22.7 ± 3.2	22.0 ± 2.9	0.35	23.1 ± 1.5	23.1 ± 2.3	0.99	0.71	0.30
PASP (mmHg)	24.5 ± 6.2	24.0 ± 5.4	0.69	25.5 ± 2.3	26.7 ± 3.5	0.25	0.59	0.18
RVAT (msec)	123.3 ± 3.3	122.2 ± 3.9	0.39	121.6 ± 5.9	121.0 ± 7.6	0.78	0.39	0.62
RVFWLS (%)	−20.9 ± 1.3	−22.3 ± 2.3	0.03	−21.0 ± 3.4	−20.7 ± 2.5	0.85	0.96	0.04
RVFWLSbas (%)	−22.2 ± 1.7	−25.2 ± 2.0	0.001	−21.6 ± 4.1	−21.6 ± 2.2	0.99	0.58	0.001
RVFWLSmid (%)	−21.7 ± 1.5	−22.5 ± 2.1	0.12	−21.3 ± 3.7	−20.6 ± 2.7	0.52	0.66	0.06
RVFWLSap (%)	−18.9 ± 2.9	−19.1 ± 3.7	0.85	−20.1 ± 3.0	−20.3 ± 4.0	0.91	0.36	0.43
RV4CLS (%)	−20.7 ± 1.1	−22.1 ± 2.6	0.03	−20.2 ± 2.7	−20.5 ± 2.7	0.72	0.45	0.05

RVbas: RV basal diameter, RVOT prox: proximal RV outflow track diameter from PLAX, RAVol: right atrial volume, TVE: early diastolic transtricuspid flow velocity, TVA: late diastolic transtricuspid flow velocity, TVE/A: ratio of early to late diastolic transtricuspid flow velocity, TVS’: systolic tricuspid annular velocity, TVE’: early diastolic tricuspid annular velocity, TVA’: late diastolic tricuspid annular velocity, TAPSE: tricuspid annular plane systolic excursion, PASP: pulmonary artery systolic pressure, RVAT: RV acceleration time, RVFWSL: RV free-wall longitudinal systolic strain, RV4CSL: RV four–chamber longitudinal systolic strain, RVFWLS bas: longitudinal systolic strain from the basal segment of the RV free wall, RVFWLS mid: longitudinal systolic strain from the middle segment of the RV free wall, RVFWLS ap: longitudinal systolic strain from the apical segment of the RV free wall.

## Data Availability

Data can be provided upon request.

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
