# Peer review of "Right Ventricular Morphology and Function after Exercise Training in People with Systemic Sclerosis: A Randomized Controlled Pilot Study"

_life, 2023, doi:10.3390/life13020545_

Round 1

Reviewer 1 Report

The authors of this manuscript report on a small, randomized prospective clinical trial investigation the effects of a 3-month program of exercise training on cardiorespiratory function using CPET and both conventional and speckle echocardiographic imaging in a cohort of subjects with SSc. While most of the parameters evaluated at baseline were relatively normal at baseline in the total cohort, there were abnormalities in left and right ventricular strain patterns that were felt to represent subclinical cardiac dysfunction. while there were no changes in any parameters in the control group at 3 months, there were improvements in VO2max and wall strain in the exercise training group. The authors conclude that these findings suggest that such an approach may identify abnormalities in SSc even prior to the onset of symptoms and that exercise training may improve cardiopulmonary function at this stage of the disease course.

Overall, the manuscript is well-written, although there are a few confusing sentences (see Abstract and Introduction in particular). The data are clearly presented, although the Discussion would benefit from greater focus and elaboration of their findings: Specifically, the echo changes seem relatively minor relative to the substantial improvement in VO2max in the exercise group. The explanation for this remains unclear to me. While the authors speculate that attenuation of pulmonary artery pressure or improved microcirculatory flow to the heart may explain these findings, they provide no evidence for these possibilities. While resting cardiac output may be normal, it is possible that the blunted increase in cardiac output with exercise that is observed in patients with PAH, even early in the course of their illness, may have improved with exercise training and explain the improved VO2max, no measurements of cardiac output or estimated RVsys with exercise are presented. One is left with wondering whether the changes in load/strain with training are meaningful if, despite a correlation with changes in VO2max, are the cause of the changes in VO2max and if they are, therefore, clinically meaningful.

The study involves a small cohort from a larger ongoing trial, and that raises the question as to whether this subgroup presentation and analysis is premature. Additionally, there are some imbalances between the two randomized groups, particularly the distribution of subjects with limited SSc, which appears to be higher in the control group. It is possible that those with diffuse disease are more likely to have subclinical cardiopulmonary and skeletal muscle disease and therefore this subgroup, in particular may benefit from both pre-symptomatic study and intervention. Another limitation is the short duration of the study protocol: It is unclear whether these improvements persist or even amplified with a longer duration of treatment and observation. Finally, the decline in VO2max after only three months in the control group, despite the absence of clear changes in echo parameters, is confusing and not addressed.

Author Response

Point to point answers to reviewer #1

The authors of this manuscript report on a small, randomized prospective clinical trial investigation the effects of a 3-month program of exercise training on cardiorespiratory function using CPET and both conventional and speckle echocardiographic imaging in a cohort of subjects with SSc. While most of the parameters evaluated at baseline were relatively normal at baseline in the total cohort, there were abnormalities in left and right ventricular strain patterns that were felt to represent subclinical cardiac dysfunction. while there were no changes in any parameters in the control group at 3 months, there were improvements in VO2max and wall strain in the exercise training group. The authors conclude that these findings suggest that such an approach may identify abnormalities in SSc even prior to the onset of symptoms and that exercise training may improve cardiopulmonary function at this stage of the disease course.

Comment: Overall, the manuscript is well-written, although there are a few confusing sentences (see Abstract and Introduction in particular). The data are clearly presented, although

Answer: The abstract and introduction have now been reviewed and appropriate corrections with tracked change have been implemented throughout.

Comment: The Discussion would benefit from greater focus and elaboration of their findings: Specifically, the echo changes seem relatively minor relative to the substantial improvement in VO2max in the exercise group. The explanation for this remains unclear to me. While the authors speculate that attenuation of pulmonary artery pressure or improved microcirculatory flow to the heart may explain these findings, they provide no evidence for these possibilities. While resting cardiac output may be normal, it is possible that the blunted increase in cardiac output with exercise that is observed in patients with PAH, even early in the course of their illness, may have improved with exercise training and explain the improved VO2max, no measurements of cardiac output or estimated RVsys with exercise are presented. One is left with wondering whether the changes in load/strain with training are meaningful if, despite a correlation with changes in VO2max, are the cause of the changes in VO2max and if they are, therefore, clinically meaningful.

Answer: We would like to thank the reviewer for this comment.

We have now included further details for i) the peripheral adaptations leading to an increase in VO2peak, ii) evidence of CO indicating no significant change as well as iii) further evidence from our previous study for the systemic benefits of exercise on the microvasculature supporting thus our hypothesis of central microvascular changes contributing to the RV function post exercise training. Please see the tracked changes in pages 10-11 lines 334-49.

Comment: The study involves a small cohort from a larger ongoing trial, and that raises the question as to whether this subgroup presentation and analysis is premature.

Answer: We would like to thank the reviewer for this comment.

The larger ongoing trial aims to assess different outcomes, which are not connected to the cardiac function and structure. This subgroup is the only group from this larger ongoing trial that have performed echocardiography pre and post exercise and thus this presentation and analysis are not considered premature.

Comment: Additionally, there are some imbalances between the two randomized groups, particularly the distribution of subjects with limited SSc, which appears to be higher in the control group. It is possible that those with diffuse disease are more likely to have subclinical cardiopulmonary and skeletal muscle disease and therefore this subgroup, in particular may benefit from both pre-symptomatic study and intervention.

Answer: We would like to thank the reviewer for this comment.

Although there is no evidence to support that those with diffuse are more prone to benefit from exercise, we have included this group imbalance as a limitation with tracked changes in page 11 Lines 376-79.

Comment: Another limitation is the short duration of the study protocol: It is unclear whether these improvements persist or even amplified with a longer duration of treatment and observation.

Answer: We would like to thank the reviewer for this comment.

This limitation has been included within the manuscript. Page 11 Lines 373-76.

Comment: Finally, the decline in VO2max after only three months in the control group, despite the absence of clear changes in echo parameters, is confusing and not addressed.

Answer: We would like to thank the reviewer for this comment.

The answer has been included within the manuscript with tracked changes in page 11 lines 357-69.

Reviewer 2 Report

The work seems to me to be of great interest to the scientific community in the field. I would only recommend improving the quality of the figures, since it is very poor.

Author Response

Thank you for your comment. We have resized the figures and change them 300dpi and 1024px width.

Reviewer 3 Report

Thank you for the opportunity to review this paper, which is designed to address an important question for the SSc population, particularly as it relates to improved cardiac dynamics post-HIT. I found the paper to be well designed and well executed, with the results easy to follow and the discussion well thought out. I do have some minor suggestions to improve the writing, which I have point-by-point below. In addition, I do have one significant suggestion about the data analysis, as there are only 4 men out of 24 total participants,  and all 4 are in intervention group (with obviously none in the control group). Therefore, I would strongly suggest that you consider removing the men entirely from the sample and re-analyzing the data with the females only. As it now stands, with so few men, and all of them in one group, it makes it quite difficult to imply whom the results apply to. I suspect you will get similar results after removing the men, but it will be much cleaner, and then you can state more definitively that the results only apply to females.

-line 55: patient should be plural

Section 2.2: Study Protocol: please describe how many were in each group, along with the gender breakdown (unless of course you follow my main suggestion of removing the men).

-line 100-101: 'the expiration gases' should be 'expiratory gases'.

line 106-07: at what point(s) during recovery was/were BP measured?

line 108: the statement that VO2 was averaged during the last 30 sec of 'each' test, implies that participants had multiple tests (more so than just the pre-post). Therefore, I might suggest stating that VO2 was recorded as the ave during the last 30 sec of 'individual's' test.

line 154: states that the arm-crank is 'the same' device. Same as what?

Section 3: Results. this sections begins with stating the number of participants, but doesn't note that the men were all in one group (again, irrelevant if you take my main suggestion).

line 237 (Discussion): 'SSc patients had diminished ...'. Diminished compared to what or whom?

line 252-53: states that SSc patients did not differ from healthy. however there was no healthy group in your study. I'm pretty sure you are talking about a previous study, so please make that clearer.

line 272-276: this brief section is not written well. Too many sentences and poor sentence structure. Please revise.

line 278: 'their' at the start of the line should be deleted.

line 284: replace 'have proven' with 'show'

line 299: replace 'that' with 'which'

Author Response

Point to point answers to reviewer #2

Thank you for the opportunity to review this paper, which is designed to address an important question for the SSc population, particularly as it relates to improved cardiac dynamics post-HIT. I found the paper to be well designed and well executed, with the results easy to follow and the discussion well thought out. I do have some minor suggestions to improve the writing, which I have point-by-point below.

Comment: In addition, I do have one significant suggestion about the data analysis, as there are only 4 men out of 24 total participants, and all 4 are in intervention group (with obviously none in the control group). Therefore, I would strongly suggest that you consider removing the men entirely from the sample and re-analyzing the data with the females only. As it now stands, with so few men, and all of them in one group, it makes it quite difficult to imply whom the results apply to. I suspect you will get similar results after removing the men, but it will be much cleaner, and then you can state more definitively that the results only apply to females.

Answer: We would like to thank the reviewer for this comment.

We have indeed excluded the men from our statistical analysis and the results did not alter significantly or with a tendency to change. Therefore, we believe and from our previous experience that the gender differences are not affecting our results. We also know that systemic sclerosis is an understudied rare rheumatoid connective tissue disease condition mainly affecting the women, with large knowledge gaps in the field of clinical exercise physiology. For this reason and following a consistent strategy (i.e., including both genders despite disease-induced gender prevalence differences) from our previous research papers in this condition, we would not like to exclude the men from this study. From an ethical perspective we believe that by excluding the men from our study we prevent the application of an effective non-pharmacological treatment to this gender (i.e., men) in future disease management guidelines.

Comment: line 55: patient should be plural

Answer: It has now been corrected with tracked changes in line 58.

Comment: Section 2.2: Study Protocol: please describe how many were in each group, along with the gender breakdown (unless of course you follow my main suggestion of removing the men).

Answer: It has now been corrected with tracked changes in line 97-99.

Comment: line 100-101: 'the expiration gases' should be 'expiratory gases'.

Answer: It has now been corrected with tracked changes in line 110.

Comment: line 106-07: at what point(s) during recovery was/were BP measured?

Answer: It has now been corrected with tracked changes in line 116.

Comment: line 108: the statement that VO2 was averaged during the last 30 sec of 'each' test, implies that participants had multiple tests (more so than just the pre-post). Therefore, I might suggest stating that VO2 was recorded as the ave during the last 30 sec of 'individual's' test.

Answer: It has now been corrected with tracked changes in line 117.

Comment: line 154: states that the arm-crank is 'the same' device. Same as what?

Answer: It has now been corrected with tracked changes in line 168.

Comment: Section 3: Results. this sections begins with stating the number of participants, but doesn't note that the men were all in one group (again, irrelevant if you take my main suggestion).

Answer: Table 1 in the results section contains all the group details so we do believe that restating this in the beginning will be a repetition. We have also included the number at the study protocol section as you suggested.

Comment: line 237 (Discussion): 'SSc patients had diminished ...'. Diminished compared to what or whom?

Answer: It has now been corrected with tracked changes in line 289-90.

Comment: line 252-53: states that SSc patients did not differ from healthy. however there was no healthy group in your study. I'm pretty sure you are talking about a previous study, so please make that clearer.

Answer: This specific sentence refers to previous studies (19-21: Matias 2009, Karadag 2020 and Kepez 2008) that had compared the strain values of the SSc with health controls.

Comment: line 272-276: this brief section is not written well. Too many sentences and poor sentence structure. Please revise.

Answer: It has now been corrected with tracked changes in line 289-95.

Comment: line 278: 'their' at the start of the line should be deleted.

Answer: It has now been corrected.

Comment: line 284: replace 'have proven' with 'show'

Answer: It has now been corrected with tracked changes in line 339.

Comment: line 299: replace 'that' with 'which'

Answer: It has been corrected with tracked changes in Line 395.

Round 2

Reviewer 1 Report

I thank the authors for their thoughtful responses and revisions to their manuscript. The discussion regarding the changes in VO2max that were observed in the two study arms enhances greatly the interpretation of their findings.

Author Response

Thank you for your comment.